# The Combined Anti-Aging Effect of Hydrolyzed Collagen Oligopeptides and Exosomes Derived from Human Umbilical Cord Mesenchymal Stem Cells on Human Skin Fibroblasts

**DOI:** 10.3390/molecules29071468

**Published:** 2024-03-26

**Authors:** Huimin Zhu, Xin Guo, Yongqing Zhang, Ajab Khan, Yinuo Pang, Huifang Song, Hong Zhao, Zhizhen Liu, Hua Qiao, Jun Xie

**Affiliations:** 1Department of Biochemistry and Molecular Biology, Shanxi Key Laboratory of Birth Defect and Cell Regeneration, MOE Key Laboratory of Coal Environmental Pathogenicity and Prevention, Shanxi Medical University, Taiyuan 030001, China; zhm18536062073@163.com (H.Z.); spuguoxin@163.com (X.G.); 13095803292@163.com (Y.Z.); 19855321606@163.com (Y.P.); songhuifang0111@yeah.net (H.S.); shanxizhaohong@163.com (H.Z.); zhizhenliu2013@163.com (Z.L.); 2Department of Veterinary Pathology, Faculty of Veterinary and Animal Sciences, The University of Agriculture, Dera Ismail Khan 29050, Pakistan; drajab22@gmail.com

**Keywords:** exosomes, hydrolyzed collagen oligopeptides, skin anti-aging, human umbilical cord mesenchymal stem cells, human skin fibroblasts, replicative senescence

## Abstract

Stem cell-derived exosomes (SC-Exos) are used as a source of regenerative medicine, but certain limitations hinder their uses. The effect of hydrolyzed collagen oligopeptides (HCOPs), a functional ingredient of SC-Exos is not widely known to the general public. We herein evaluated the combined anti-aging effects of HCOPs and exosomes derived from human umbilical cord mesenchymal stem cells (HucMSC-Exos) using a senescence model established on human skin fibroblasts (HSFs). This study discovered that cells treated with HucMSC-Exos + HCOPs enhanced their proliferative and migratory capabilities; reduced both reactive oxygen species production and senescence-associated β-galactosidase activity; augmented type I and type III collagen expression; attenuated the expression of matrix-degrading metalloproteinases (MMP-1, MMP-3, and MMP-9), interleukin 1 beta (IL-1β), and tumor necrosis factor-alpha (TNF-α); and decreased the expression of p16, p21, and p53 as compared with the cells treated with HucMSC-Exos or HCOPs alone. These results suggest a possible strategy for enhancing the skin anti-aging ability of HucMSC-Exos with HCOPs.

## 1. Introduction

The aging of the skin is associated with both extrinsic and intrinsic factors that cause the aging of skin phenotypes, including wrinkles, skin dryness, irregular pigmentation, and a decrease in dermal and epidermal thickness [1]. With aging, degenerative changes occur in the structure of the skin, with the most obvious changes being seen in the dermis [2]. The dermis is located between the subcutaneous and epidermal layer and possesses a network of dense extracellular matrix, thus giving mechanical support and elastic recoil to the skin [1]. The primary cell types present in the dermis are fibroblasts, which are capable of producing structural components, including collagen, the major part of ECM in the dermis [3]. With aging, the capability of fibroblast proliferation and collagen synthesis is gradually lost [4]. Both extrinsic and intrinsic factors involved in aging reduce the proliferation rate of dermal fibroblasts, which results in a decrease in collagen production, enhances the breakdown of matrix-degrading metalloproteinases (MMPs), and thus induces wrinkle formation [5].

Today’s research has been focused on using stem cell therapy for skin regeneration and as an agent to stop aging. Both preclinical and clinical studies have shown that mesenchymal stem cell (MSC) transplantation helps in wound repair and facial rejuvenation [6,7]. Apart from recent achievements, the clinical application of stem cell therapy faces various challenges, including the low survival rate of stem cells after their transplantation [8]. Consequently, better methods need to be found to adjust stem cell application in the clinical setting. Clinical studies have promisingly shown the paracrine properties of transplanted stem cells, particularly by exosomes, playing a crucial role in the treatment and changes that occur in damaged tissues [9]. Exosomes are extracellular vesicles with an endosomal origin, ranging from ~40 to 160 nm in diameter, which are filled with bioactive substances (nucleic acids, cytokines, proteins, and other molecules) [10]. These bioactive compounds are transported by exosomes, which interact with specific recipient cells by initiating downstream signals or cell-to-cell communication [11]. Therefore, considerable interest has been elicited in the clinical application of exosomes. Numerous studies have recently shown the therapeutic role of MSC-derived exosomes to regenerate skin and enhance various renal, cardiovascular, and liver injuries [12,13,14], as well as have therapeutic potential in skin anti-aging. However, low yield, difficult preparation, and harsh transportation and storage conditions lead to the partial degradation of active components in exosomes and thus decline their anti-aging ability to limit the wide application of exosomes [15]. In some studies, exosomes are combined with drugs or specific therapeutic components to improve disease treatment [16,17,18], providing a method to enhance the applicative value of exosomes.

Therefore, after preliminary screening, we compared the efficacy of typical anti-aging drugs such as collagen peptides, ascorbic acid, VB6, rice protein, rapamycin, and resveratrol, and the possible effect of various combinations of these drugs with exosomes was investigated. Based on the pre-experimental results, the most therapeutically promising HCOPs were ultimately identified as the combined component of exosomes with HCOPs, which was selected for further investigation of their anti-aging efficacy. Hydrolyzed collagen oligopeptides (HCOPs) are enzyme hydrolytic forms of collagen, consisting of a popular ingredient that is supposed to be an antioxidant and thus have anti-aging effects on the skin [19]. Collagen hydrolysate is a cosmetic ingredient that is safely used as a moisturizing agent in topical formulations to reduce skin aging, including laxity, dryness, and wrinkle formation [20]. Chotphruethipong et al. [21] showed that hydrolyzed collagen, when combined with vitamin C, increases fibroblast proliferation and migration abilities, thus enhancing wound healing. Therefore, this study hypothesized that HCOPs combined with exosomes could also promote an anti-senescence effect on human skin fibroblasts (HSFs). No such information has been found concerning the synergistic effects of HCOPs and exosomes for the treatment of the replicative senescence of HSFs.

This study isolated exosomes from a human umbilical cord mesenchymal stem cell-conditioned medium (HucMSC-CM), and the combination effects of HCOPs and human umbilical cord mesenchymal stem cell-derived exosomes (HucMSC-Exos) on senescent HSFs were elucidated in vitro. To induce natural senescence, HSFs were subcultured and serially passaged. This study also revealed aging biomarkers at the protein level as well as changes in the commensurate signaling pathways. Our results have provided experimental and theoretical bases for the application of HCOPs on HucMSC-Exos in clinical anti-aging.

## 2. Results

### 2.1. Characterization of Exosomes

The nanoparticle tracking analysis (NTA) results revealed that the size distribution of HucMSC-Exos showed a key peak at ~130.5 nm with a mean diameter of 141 nm (Figure 1A). TEM images indicated a size (127.7 nm) parallel to that of the NTA results, which showed that HucMSC-Exos had a cup-shaped circular membrane structure (Figure 1B). A BCA assay showed protein concentration of HucMSC-Exos obtained from 100 mL HucMSC cm, which was 2149 µg/mL. Immunoblot analysis revealed that HucMSC-Exos possessed the exosomal markers CD81 and ALIX (Figure 1C). Combined incubation of HSFs with PKH26^®^-labeled exosomes confirmed the uptake of exosomes by the cells (Figure 1D).

### 2.2. Establishment of a Model of Replicative Senescence of HSFs

Replicative cellular senescence was induced when HSFs were cultured for a prolonged period with serial passaging. At passage 20, the induction of replicative senescence in senescent cells was confirmed through examination of cellular proliferation and morphology, cell cycle kinetics, senescence-associated β-galactosidase (SA-β-Gal) activity, and senescence-associated heterochromatin foci (SAHF) compared to young cells at passage 3. Cell proliferation showed a decline in cell growth during the process of replicative senescence (Figure 2A). Compared with P3 cells, P20 cell proliferation was significantly inhibited (*p* < 0.001). The young cells in the P3 group were spindle-shaped and compact with clear cell boundaries; while in the P20 group, cells showed an irregular arrangement, hypertrophy, flat morphology with an appearance of more particles in the cytoplasm, and unclear cell boundaries (Figure 2B). HSF cells at passages 3 and 20 were subjected to SA-β-gal staining (a reliable marker of cellular senescence) in vitro [22]. In comparison with passage 3 (0.05 ± 0.01%) (*p* < 0.001), a significant increase (%) was observed in the SA-β-gal-positive area at passage 20 (1.06 ± 0.18%) (Figure 2C). Another key feature of cellular senescence is cell cycle arrest, this study employed flow cytometry to know the distribution of cell cycle in senescent cells. Compared with P3 cells, the proportion of P20 cells in the G1 phase was increased from 36.3% to 53.8%, showing a significant G1 phase arrest (Figure 2D and Appendix A). Furthermore, SAHF formation in senescent cells was evaluated by immunofluorescence, which showed an extensive change in chromatin structure often accompanied by cellular senescence, appearing as a dot-like heterochromatin structure. As shown in Figure 2E, cells at passage 20 were found to exhibit the SAHF-aggregative phenomenon. These results showed that in passage 20, senescence was induced in cells.

### 2.3. Effect of Different Concentrations of HucMSC-Exos or HCOPs on the Proliferation of Senescent HSFs

The effect of various concentrations of HucMSC-Exos on the proliferation of HSFs is shown in Figure 3A. None of the treated HucMSC-Exos exerted any cytotoxic effects; instead, they stimulated the growth of HSFs. Among the levels of HucMSC-Exos, HucMSC-Exos-treated cells at 10 µg/mL (141.00% ± 1.28%) (*p* < 0.05) showed the highest cell proliferation, while lower proliferation was observed when HucMSC-Exos ranging from 2.5 µg/mL to 5 µg/mL and up to 20 µg/mL (114.23% ± 2.20% to 115.66% ± 1.87% and 111.60% ± 2.63%) (*p* > 0.05) were used.

As shown in Figure 3B, the effectiveness of HCOPs, especially at 0.1 µg/mL, promoted cell proliferation more effectively than the control (138.66% ± 0.75%) (*p* < 0.05). The use of HCOPs ranging from 0.1 to 10 µg/mL had a similar effect on cellular proliferation (128.16% ± 2.23% and 128.83% ± 0.87% to 125.73% ± 1.97%, respectively) (*p* > 0.05), as shown by 0.01 µg/mL. Compared with other treatment groups, 0.1 µg/mL had the strongest effect, as shown in Figure 3B. As HucMSC-Exos at 10 µg/mL and HCOPs at 0.1 µg/mL showed the maximum efficacy on cell proliferation and were therefore selected for further studies.

### 2.4. Combined Effect of HCOPs and HucMSC-Exos at the Selected Concentrations on the Proliferation and Migration of HSFs

The effects of HucMSC-Exos, HCOPs, and HucMSC-Exos supplemented with HCOPs (HucMSC-Exos + HCOPs) were investigated at the selected concentrations on both the division and migration of HSFs in vitro. Compared with the control group, all the treated groups healed significantly at a faster rate (*p* < 0.05), as shown in Figure 4A,B. The efficacy of HucMSC-Exos-treated cells was higher than that of HCOP-treated cells at 24 h (*p* < 0.05) and was the same as that of HCOP-treated cells at 48 h (*p* > 0.05). However, an increased effect was observed on cell migration for cells treated with HucMSC-Exos + HCOPs compared to the above two groups both at 24 h and 48 h (*p* < 0.05). All the treated samples showed an increased level of cell proliferation than the control (*p* < 0.05), as shown in Figure 4C. HucMSC-Exos-treated cells and HCOP-treated cells at their optimal levels showed no significant difference (*p* > 0.05), suggesting that the proficiency in inducing HSF proliferation of HucMSC-Exos at 10 µg/mL was similar to that of HCOPs at 0.1 µg/mL. Combined HucMSC-Exos + HCOPs showed higher cell proliferation (168.73% ± 2.74%) compared to HucMSC-Exos (141.00% ± 3.65%) or HCOPs (135.67% ± 8.51%) alone (*p* < 0.05).

### 2.5. Combined Effect of HCOPs and HucMSC-Exos at the Selected Concentrations on the Expression of ROS and SA-β-Gal Activity in HSFs

According to the free radical theory of aging, the leading cause of functional decline is oxidative damage caused by ROS, which is a characteristic of aging [23]. Compared with the control group, all treated samples down-regulated the level of ROS production (*p* < 0.05) (Figure 5A,B). Among the treatments, a greater diminution in ROS level was found for the HucMSC-Exos + HCOPs treatment group (24.93% ± 1.42%, *p* < 0.05), while the HucMSC-Exos and HCOPs treatment groups showed less decrease (44.61% ± 4.96%; 47.29% ± 3.86%, *p* < 0.05) (Table 1).

Compared with the control (*p* < 0.05), co-cultured senescent HSFs with HucMSC-Exos, HCOPs, and HucMSC-Exos + HCOPs inhibited SA-β-Gal-positivity (Figure 5C,D). When all the treatments were compared, the lowest SA-β-Gal level was found in HucMSC-Exos + HCOPs (30.05% ± 3.71%, *p* < 0.05), while HucMSC-Exos and HCOP samples showed similar lower levels (60.30% ± 7.00%, 62.76% ± 10.02%, *p* > 0.05) (Table 1).

### 2.6. Combined Effect of HCOPs and HucMSC-Exos at Selected Concentrations on ECM Construction-Related Proteins and Senescence-Associated Secretory Phenotype (SASP) in HSFs

As shown in Figure 6A,B, collagen I and III expression was increased in the supernatant of HSFs treated with hucMSC-Exos, HCOPs, and HucMSC-Exos + HCOPs when compared to the control (*p* < 0.05). All the treated samples showed the highest levels of collagen I and III in HucMSC-Exos + HCOPs (*p* < 0.05), while HucMSC-Exos and HCOP treatments showed similar collagen levels (*p* > 0.05). By contrast, all tested samples showed a reduction in the expression of MMP-1, MMP-3, and MMP-9 when compared with the control group (*p* < 0.05), as shown in Figure 6C–E. The expression of MMPs among the treatments showed a marked decrease in MMP-1, MMP-3, and MMP-9 in HucMSC-Exos + HCOPs group (*p* < 0.05), while the HucMSC-Exos group showed a reduction as compared to the HCOPs group (*p* < 0.05).

Compared with the control, TNF-α and IL-1β levels in all the treated groups were decreased (*p* < 0.05), as shown in Figure 6F,G. When the expression levels of TNF-α and IL-1β among the treated samples were compared, the lowest of them were observed in HucMSC-Exos + HCOPs group (*p* < 0.05), followed by HucMSC-Exos group than the HCOPs group (*p* < 0.05). These results concluded that HCOPs combined with HucMSC-Exos had a synergistic effect in the reconstitution of the dermal matrix and reduction of inflammation.

### 2.7. Combined Effect of HCOPs and HucMSC-Exos at the Selected Concentrations on Senescence-Associated Regulators

Compared with the control group, cells treated with HucMSC-Exos, HCOPs, and HucMSC-Exos + HCOPs showed significantly reduced expression of p16 and p21 (Figure 7A,B, *p* < 0.05). Among the three treated samples, the lowest levels of p16 and p21 were found in the HucMSC-Exos + HCOPs group (*p* < 0.05), while no significant difference was noted between the cells treated with HucMSC-Exos and HCOPs (*p* > 0.05). The expression of p53 was significantly reduced when treated with HCOPs and HucMSC-Exos + HCOPs as compared with the control group (Figure 7C, *p* < 0.05), except HucMSC-Exos, which was similar to the control (*p* > 0.05). Moreover, HucMSC-Exos + HCOPs induced a much lower level of p53 as compared to HucMSC-Exos (*p* < 0.05). On the contrary, an increase was observed in lamin B1 expression level in cells treated with HucMSC-Exos, especially with HucMSC-Exos + HCOPs, as compared to that of the control (Figure 7D, *p* < 0.05).

The expression levels of p16, p21, and p53 proteins were analyzed by Western blotting, and the results were consistent with the gene expression levels, except for HucMSC-Exos in p53, which was lowered than that of the controls (*p* > 0.05). As shown in Figure 8A–D, the expression levels of p16, p21, and p53 proteins of all three treated groups were decreased as compared to the controls (*p* < 0.05). Furthermore, the HucMSC-Exos + HCOPs treatment was more effective when compared with the other two treatments (*p* < 0.05).

## 3. Discussion

Recent studies in the literature have shown a great deal of concern regarding skin aging and have developed strategies to postpone or even reverse the process of skin aging, which has become a topic of interest in medical cosmetology and anti-aging research [24]. In this regard, the molecular mechanisms and factors that affect the process of skin aging have been widely studied. It has been elucidated that both intrinsic and extrinsic factors are involved in the aging of skin [20]. Intrinsic factors are related to genetics, which occurs inevitably with age. However, studies have shown that changes related to epigenetic and post-translational mechanisms are essential determinants of intrinsic aging than genetics. Extrinsic aging is simply related to external factors including improper nutrition, smoking, air pollution, and exposure to ultraviolet rays [3]. The literature has shown the protective effect of exosomes or hydrolyzed collagen on skin photoaging [25,26,27]. However, there are few studies on exosomes or hydrolyzed collagen delaying the natural aging of the skin. Furthermore, based on the review of the literature, no study has been found related to HCOPs combined with HucMSC-Exos to enhance the anti-aging effect on the replicative senescence of HSFs. As the name indicates, replicative senescence is to stop the process of cell division accompanying replicative exhaustion and is a commonly used experimental aging model based on the intrinsic mechanisms of organ aging [28]. Zhou [29] showed that exosomes from umbilical cord MSCs overexpressing NAMPT promoted proliferation and delayed the aging of senescent human skin fibroblasts (at passages 20–25) and enhanced the antioxidant capacity of Caenorhabditis elegans. This study established a replicative cellular senescence model through serial passages of normal diploid HSFs in vitro. At passage 20, cellular senescence induction in aged HSF cells was confirmed by SA-β-Gal staining Figure 2B). In addition, changes associated with cellular senescence progression, including a decrease in proliferation, cell cycle arrest, and flattened morphology, as well as SAHF were observed in senescent cells (Figure 2). Thus, subsequent anti-senescence intervention experiments were developed based on this replicative senescence model.

The application of human umbilical cord mesenchymal stem cells (HucMSCs), a transplantable source of MSCs is of great interest in skin regeneration and anti-aging due to their low cost, ease to isolate, minimal invasiveness, low immunogenic and immunomodulatory activity, and differentiation multipotency [30,31]. Moreover, advances in HucMSC transplantation demonstrate their great potential for treating skin aging; still, their long-term safety is a matter of high concern [32]. Fortunately, various researches have shown that paracrine functional components from stem cells have potential therapeutic effects on various pathological conditions, of which exosomes are considered crucial players [33]. Due to lower immune responses, high safety level, reliability of healthy skin preservation, and ease of administration, HucMSC-Exos therapies are preferred when compared with MSC transplantation [13]. Lei et al. [34] found that exosomes derived from umbilical cord mesenchymal stem cells reverse the aging of adult bone marrow MSCs and significantly enhance osteogenesis and the repair ability of the latter, showing the treatment potential of HucMSC-Exos on replicative senescent fibroblasts.

HCOPs are known antioxidants for the treatment of skin aging, thus enabling them to be used in food supplementation and cosmetic skincare because of their better solubility and bioavailability with low levels of allergic properties when compared with traditional collagen formulations [35]. Peptides derived from marine protein hydrolysates confirm that peptides derived from fish proteins have higher antioxidant properties compared with the natural antioxidant, a-tocopherol, in different oxidative systems [36]. León-López et al. demonstrated that the antioxidant properties of the hydrolyzed collagen were dependent on the low molecular weight of peptides, because smaller molecules having an average molecular weight of 5 kDa had a higher ability to contribute an electron or hydrogen to stabilize oxygen radicals than larger molecules [37]. Song et al. [38] revealed that, compared to larger peptides, smaller ones generally have greater bioactivity, especially to activate fibroblast proliferation and migration, and are readily absorbed in the human body [39]. The hydrolyzed collagen oligopeptides in this study were made from the skin of deep-sea salmon after cleaning, compound-coupling enzymatic digestion, enzyme deactivation, degreasing, multistage membrane separation, permeation and concentration, decolorization, drying, and other processes. The molecular mass range was principally between 100 and 860 Da. These results suggest that the HCOPs in our study may possess stronger biological activity in antioxidant and anti-aging aspects. Along with the size, the composition of amino acids also plays a vital role in its bioactivities. The HCOPs’ amino acid composition (Appendix A) indicated that a high level of hydrophobic amino acids were present in HCOPs obtained from salmon skin (53.8%), which is related to various biological activities of peptides, including antioxidant potency and fibroblast division [40]. In addition, proline, glycine, and alanine are predominantly hydrophobic amino acids, which likely promote skin cell viability [41]. To enhance the anti-aging effect on skin, HCOPs have been combined with some active ingredients such as hyaluronic acid, vitamins, and elastin peptides, and enhanced results were obtained for these drugs [21,42]. Thus, we hypothesize that HCOPs combined with HucMSC-Exos enhance the anti-aging effect on the replicative senescence of HSFs.

In this study, exosomes were first purified from the medium conditioned by HucMSCs and characterized as HucMSC-Exos by having a cup-shaped circular membrane structure and being approximately 40 to 160 nm in diameter (Figure 1). Exosomes derived from other cell lines also showed the same features [43]. Shabbir et al. [44] showed that after entering into the fibroblasts, MSC exosomes primarily gather around the nucleus, which was confirmed by the result of this study in in vitro exosome uptake experiments (Figure 1D). When HucMSC-Exos were added to HSF cells (Figure 3A), they accelerated cellular proliferation that may be caused by various biomolecules in exosomes, including fibrillin, ankyrin, desmin, vimentin, α-2-macroglobulin, and fibronectin involved in tissue repair [31,45]. However, our results did not show that the higher the contents of HucMSC-Exos, the greater the cellular proliferation of HSFs, as shown by Oh et al. [3]. The optimal concentration of HucMSC-Exos was 10 µg/mL, with a 41% increase in HSF proliferation (Figure 3A). When HCOPs (0.1 µg/mL) were deployed into HucMSC-Exos, we found a much higher cellular proliferative activity, with an increase of up to 68.7% (Figure 4A). This indicated that HCOPs combined with HucMSC-Exos effectively enhanced the proliferation of HSFs. HSFs were used to check the impact of HucMSC-Exos + HCOPs on cell migration, which confirmed greater efficacy on wound healing compared to HucMSC-Exos alone (Figure 4B). The reason may be the higher contents of hydrophobic amino acids proline and arginine found in HCOPs, which were documented to induce fibroblast proliferation and migration [41,46] (Appendix A). These data implied that HCOPs combined with HucMSC-Exos enhanced the effect in improving cell viability and migration, and our results supported this hypothesis.

Next, we further demonstrated the combined effect of HCOPs and HucMSC-Exos for multiple genotypic changes involved in natural senescence. The expression levels of ROS and SA-β-Gal are age-related, and we ascertained that the highest effectiveness in reducing ROS and SA-β-Gal levels occurred when cells treated with HucMSC-Exos were supplemented with HCOPs (Figure 5), suggesting that HCOPs combined with HucMSC-Exos potentiated the effect of rejuvenating senescent HSFs. Furthermore, HucMSC-Exos + HCOPs treatment more effectively enhances the expression levels of collagen type I and III with a more robust reduction in the expression of MMP-1, MMP-3, and MMP-9 in senescent HSFs when compared to other treatments (*p* < 0.05) (Figure 6 and Appendix A). These results suggest that HucMSC-Exos contained useful factors that induce the expression of genes involved in aging and thus promote the reconstruction of a dermal matrix by aggregating the content of structural proteins, including collagen type I in aged skin, similar to the results of Guo et al. [11]; and HCOP incorporation enhanced this process. Furthermore, it was also confirmed that HucMSC-Exos inhibited IL-1β and TNF-α in senescent HSFs (Figure 6 and Appendix A), similar to the result of Hu et al. [5], while HCOPs enhanced this negative effect. These results concluded that HucMSC-Exos treatment efficacy was partially related to reducing inflammation and suppressing MMPs, which can be greatly augmented by HCOPs.

The two crucial signaling pathways involved in the process of cellular senescence are P16/Rb and p19/p53/p21 [47]. P16 is a vital tumor suppressor gene that has a critical role in the regulation of the cycle by preventing cells from the G1 phase from transitioning and entering into the S phase. The accumulation of tissue aging and cell-division times up-regulate the expression of p16 [48], which is consistent with our results, in which P20 cells showed significant G1 phase arrest (Figure 1D) compared with P3 cells. Moreover, p21 and p53 are also crucial genes that have a role in cell senescence [49]. When cell senescence and DNA damage occur, p53 is activated, which induces a gradual increase in the expression of its protein products during the process of cellular senescence. One of its downstream effector proteins is p21, and its level of expression is also elevated in senescent cells. This study demonstrated greater effectiveness in down-regulating genes and protein expression levels of p53, p21, and p16 when HucMSC-Exos supplemented with HCOPs were treated (Figure 7 and Figure 8 and Appendix A). Notably, treatment with HucMSC-Exos showed small differences in the expression of the p53 gene relative to the control (*p* > 0.05); however, HCOP incorporation strengthened the regulatory effect of HucMSC-Exos. These results indicated that HCOPs combined with HucMSC-Exos enhanced the effect in down-regulating cell cycle regulators.

This study elucidated how HCOPs combined with HucMSC-Exos markedly enhanced the anti-senescence effects on senescent HSFs in vitro by promoting cellular proliferation and migration and delaying cellular senescence by reducing the expression levels of ROS, MMPs, IL-1β, and TNF-α. These actions might be mediated by anti-oxygenation, ECM reconstruction, and reduced inflammation induced by HucMSC-Exos + HCOPs. However, as many complex processes are being involved in skin aging, further investigations are needed to elucidate the mechanism(s) underlying the combined effects of HCOPs and HucMSC-Exos. Additional aspects must be considered when facing the utilization of exosomes in aesthetic medicine since processing procedures and techniques (grafting, isolation, purification, optimization, administration) are not fully agreed upon [50].

## 4. Materials and Methods

### 4.1. Cell Culture

HucMSCs (provided by Prof. Huifang Song, Shanxi Medical University, Taiyuan, China) and HSFs (Otwo Biotech, Shenzhen, China) were placed in Dulbecco’s modified Eagle medium/nutrient mixture F12 (DMEM/F12; Gibco, Grand Island, NY, USA) supplemented with 10% and 20% fetal bovine serum (FBS; Gibco, Grand Island, NY, USA), respectively, with 100 U/mL penicillin and 100 μg/mL streptomycin. Replicative senescence was induced by culturing HSFs for a long time and passaged repeatedly, as described by Oh et al. and Zhou et al. [3,29]. All cells were cultured at 37 °C in a humidified atmosphere with 5% CO_2_.

### 4.2. Exosome Isolation

FBS depleted with extracellular vesicles were prepared through ultracentrifugation at 100,000× *g* for 18 h at 4 °C, followed by filtering (0.22-µm filter, MilliporeSigma, Burlington, MA, USA). When the HucMSCs were 60–70% confluent, the culture medium was replaced with a fresh medium supplemented with EV-depleted 10% FBS. Post incubation of 48 h, HucMSC-CM were collected and centrifuged (800× *g* for 20 min). Then, the supernatant was collected, and it was centrifuged for another 30 min at 10,000× *g*. The larger particles in the supernatants (greater than 200 nm) were sieved (0.22-µm filter, MilliporeSigma, Burlington, MA, USA). Lastly, exosomes were sequestered by ultracentrifugation (120,000× *g* for 70 min), and the pellet formed was consequently washed with phosphate-buffered saline (PBS) subjected to ultracentrifugation. The re-suspended exosomal pellet was estimated using a BCA assay for protein concentration. The aliquots were then stored at −80 °C until further use.

### 4.3. Characterization of HucMSC-Exos

Zeta-View (Particle Metrix, Inning am Ammersee, Germany) was used to evaluate the concentration and size of exosomes, while a transmission electron microscope (TEM, Hitachi, HT7800, Tokyo, Japan) was used to record their morphology. Briefly, the exosomal suspension was absorbed on a copper grid with carbon film (3–5 min) and stained with 2% phosphotungstic acid for 1–2 min. Post absorption of excess liquid with filter paper, HucMSC-Exos were visualized using TEM, and images were captured. The exosomal marker proteins CD81 and ALIX (Proteintech, Rosemont, IL, USA) were analyzed by Western blotting.

### 4.4. In Vitro Exosome Uptake Assay

For the verification of exosome internalization into skin cells, HSF cells (7 × 10^4^ cells per well) were inoculated using 24-well culture plates. In accordance with the manufacturer’s instructions, post 24 h of incubation, a PKH26^®^ red fluorescent cell linker kit (Solarbio, Beijing, China) was used to label exosomes from HucMSCs. A total of 10 µg/mL of labeled exosomes were co-cultured with HSFs for 12 h, with 4,6-diamidino-2-phenylindole (DAPI) for staining nuclei. Zeiss LSM 780 confocal microscopy system (Carl Zeiss Meditec AG, Jena, Germany) was used for capturing photographs.

### 4.5. Proliferation Assay

HSF proliferation assays were performed using a Cell Counting Kit-8 (CCK-8, Bioss, Zhuhai, China). HSFs were inoculated in a 96-well plate in a ratio of 5000 cells per well and incubated for 24 h in a growth medium. Following treatment with HucMSC-Exos or HCOPs (obtained from China Research Institute of Daily Chemical Industry) in DMEM/F12 supplemented with 5% FBS for 48 h, CCK-8 reagent was supplemented to each well, followed by 2 h of incubation. A microplate spectrophotometer (Molecular Devices, SpectraMax 190, San Jose, CA, USA) was used at 450 nm to read and record the absorbance. To elucidate the enhancement impact of HCOPs on HucMSC-Exos, we applied HucMSC-Exos and HCOPs at the selected concentrations.

### 4.6. Cell Cycle Assay

HSF cells were seeded (3 × 10^5^ cells/well) into six-well cell culture plates. After 80% confluency, the cells were digested and collected using 0.25% trypsin solution and fixed with the help of a 70% pre-cooled ethanol solution (overnight at 4 °C). Using the manufacturer’s instructions, HSFs were stained with PI and examined with a FACS Canto flow cytometer (BD Biosciences, Franklin Lakes, NJ, USA).

### 4.7. Migration Assay

HSF cells (3.5 × 10^5^ cells/well) were seeded in six-well plates, and a sterilized tip was used to scratch a wound. The selected concentrations of HucMSC-Exos, HCOPs, and HucMSC-Exos + HCOPs were added to the scratched cells and incubated for 24 and 48 h. An inverted microscope (Nikon, Ti2-U, Tokyo, Japan) was used to photograph the wound gaps, and the gaps were calculated with ImageJ 2.0 software (NIH, Bethesda, MD, USA). The results are expressed as % of scratch closure in comparison to those found at 0 h.

### 4.8. SA-β-Gal Staining

Senescent HSFs were inoculated in 24-well plates at a ratio of 3.5 × 10^4^ cells/well and incubated for 24 h. Following treatment with HucMSC-Exos, HCOPs, or HucMSC-Exos + HCOPs for 48 h, cells were fixed with 4% paraformaldehyde, senescent cell histochemical staining kit (Beyotime Biotechnology, Shanghai, China) was used to stain the SA-β-Gal. Three photographs per well were captured using an inverted microscope (Nikon, Ti2-U, Japan), and the cells stained with SA-β-Gal were counted.

### 4.9. ROS Content

Intracellular ROS levels were investigated using an ROS assay kit (Beyotime Biotechnology, China). HSFs (3 × 10^4^ cells/well) were seeded in 24-well plates for 24 h of incubation. Post 48 h of treatment with HucMSC-Exos and HCOPs, the cells were incubated with 2′-7′-dichlorofluorescein diacetate for 20 min at 37 °C. The HSF ROS content was detected using an inverted fluorescence microscope (Nikon, Ti2-U, Japan) for excitation at a 488 nm wavelength and emission at a 525 nm wavelength.

### 4.10. SAHF Assay

Immunofluorescence was used to analyze SAHF formation in cultured cells. Briefly, HSF cells were seeded (5 × 10^4^ cells/well) into sterile cell climbing slices. After 80% confluency, HSFs were fixed with 4% paraformaldehyde and permeabilized with 0.5% Triton-X-100 solution at room temperature. Post three times washing with PBS, PBS containing 1% BSA was used to block the cells for 1 h. The prepared primary antibody specific for K9M-H3 was added, and the complex was incubated overnight at 4 °C. Secondary antibodies specific for DyLight 550 Conjugated AffiniPure goat anti-rabbit IgG (H+L) (BOSTER Biological Technology, Wuhan, China) were added with a 1 µg/mL DAPI (Solarbio, Beijing, China) solution for nuclei staining. A confocal microscope (Leica, Wetzlar, Germany) was used to capture the photographs. Cells with condensed K9M-H3 that co-localized with DAPI in the nuclei were considered to be SAHF-positive, and SAHF-positive cell percentages were calculated relative to the total number of cells.

### 4.11. Enzyme-Linked Immunosorbent Assay (ELISA)

The concentrations of Type I and Type III collagen, matrix metalloproteinases including MMP-1, MMP-3, and MMP-9, tumor necrosis factor-alpha (TNF-α), and interleukin 1 beta (IL-1β) were measured by ELISA (Abcam, Cambridge, UK). HSFs were first seeded in six-well plates and incubated for 24 h. Post treatment of HucMSC-Exos, HCOPs, or HucMSC-Exos + HCOPs in co-culture for 48 h, the conditioned medium was collected, and type I and type III collagen, MMP-1, MMP-3, MMP-9, TNF-α, and IL-1β levels were quantified using respective ELISA kits. A microplate spectrophotometer (Molecular Devices, SpectraMax 190, USA) was used to measure the absorbance at 450 nm.

### 4.12. Quantitative Real-Time Reverse-Transcription Polymerase Chain Reaction (qRT-PCR)

Following the manufacturer’s guidelines, TransZol Up Plus RNA Kit (TransGen Biotech, Beijing, China) and TransScript^®^ IV One-Step gDNA Removal and cDNA Synthesis SuperMix (TransGen Biotech, China) were used to extract total RNA from HSFs and reverse-transcribe cDNA, respectively. Using specific primers and 2X M5 HiPer Realtime PCR Super Mix with Low Rox (SYBR green with anti-Taq, Mei5Biotechnology, Beijing, China) in triplicate, an Eco Real-Time PCR System (Illumina, San Diego, CA, USA) was used to conduct PCR on the cDNA samples. The primers used are shown in Appendix A. Relative mRNA expression levels were normalized to GAPDH and calculated using the 2^−∆∆Ct^ method with standard cycle conditions set as 95 °C for 1 min, followed by 40 cycles of denaturation at 95 °C for 15 s, annealing at 60 °C for 15 s, and extension at 72 °C for 1 min.

### 4.13. Western Blotting Analysis

RIPA buffer (Epizyme Biotech, Shanghai, China), having protease inhibitors, was added to four HSFs groups, and an ultrasonic cell disruptor was applied to lyse the cells. Following centrifugation, the pellet was discarded, and the protein concentration was determined using a BCA protein assay kit (Beyotime Biotechnology, China). Proteins were separated using 10–12% SDS-polyacrylamide gel with 20–30 µg protein samples. The proteins were then transferred to polyvinylidene fluoride membranes (Millipore, St. Louis, MO, USA). Blots were incubated with each primary antibody of target proteins (Rabbit polyclonal anti-CD81, Rabbit polyclonal anti-ALIX, Rabbit polyclonal anti-CDKN2A/p16-INK4a, Rabbit polyclonal anti-p21 Cip1, Mouse monoclonal anti-p53). Post washing with TBST, the membrane was incubated again with a secondary antibody of goat anti-rabbit IgG and goat anti-mouse IgG conjugated to HRP (1:5000). Targeted protein bands were then scanned using an ImageQuant LAS 4000 Chemiluminescence Imager (GE, Salt Lake, UT, USA). Taking GAPDH as the reference protein, Image J 2.0 software was used to analyze the gray value to quantify protein expression.

### 4.14. Statistical Analysis

All the results expressed as mean ± standard deviation (SD) were statistically analyzed using Student’s *t*-test and one-way ANOVA using IBM SPSS Statistics 26 software. *p* < 0.05 was statistically considered significant.

## 5. Conclusions

In summary, our results revealed a combined effect of HCOPs and HucMSC-Exos in the treatment of the natural senescence of HSFs. To the best of our knowledge, this is the first ever study in which the synergistic effects of HCOPs from salmon fish skin were investigated on HucMSC-Exos for the treatment of aging skin. We demonstrated that HCOPs, as functional ingredients or nutraceuticals for skin nourishment and skin anti-aging, greatly bolstered the skin anti-aging effects of exosomes. Although the mechanism(s) underlying this enhancing action necessitates further elucidation, this study provides a technical foundation for improving the therapeutical efficacy of HucMSC-Exos in skin aging.

## Figures and Tables

**Figure 1 molecules-29-01468-f001:**
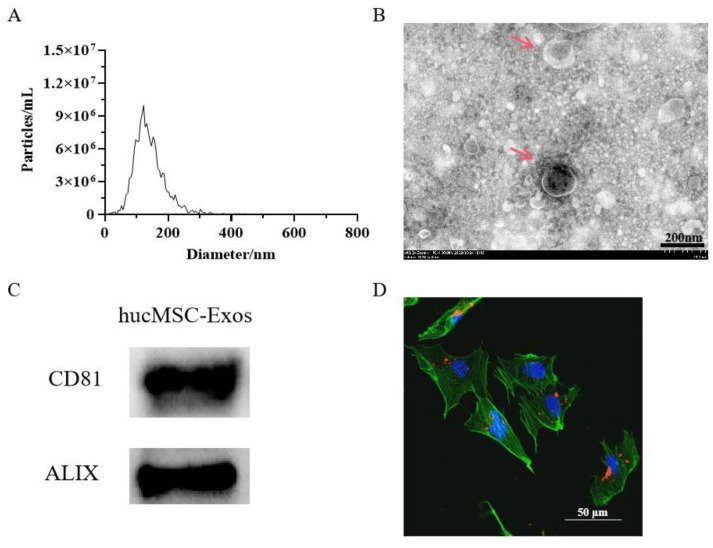
Characterization of exosomes derived from human umbilical cord mesenchymal stem cells (HucMSC-Exos). (**A**) Nanoparticle tracking analysis (NTA) of HucMSC-Exos. The mean diameter of HucMSC-Exos was 141 nm. (**B**) Transmission electron microscopic (TEM) analysis of exosomes; The red arrows point to the HucMSC-Exos; scale bars are 200 nm. (**C**) Immunoblotting for CD81 and ALIX in exosomes. (**D**) Verification of the uptake of exosomes in skin cells. HucMSC-Exos were stained with PKH26^®^ (red) and incubated with human skin fibroblasts (HSFs) for 12 h. Before analysis, cells were counterstained with phalloidin (green) and nuclei were stained with 4,6-diamidino-2-phenylindole (DAPI) (blue) for counterstaining; scale bars are 50 µm.

**Figure 2 molecules-29-01468-f002:**
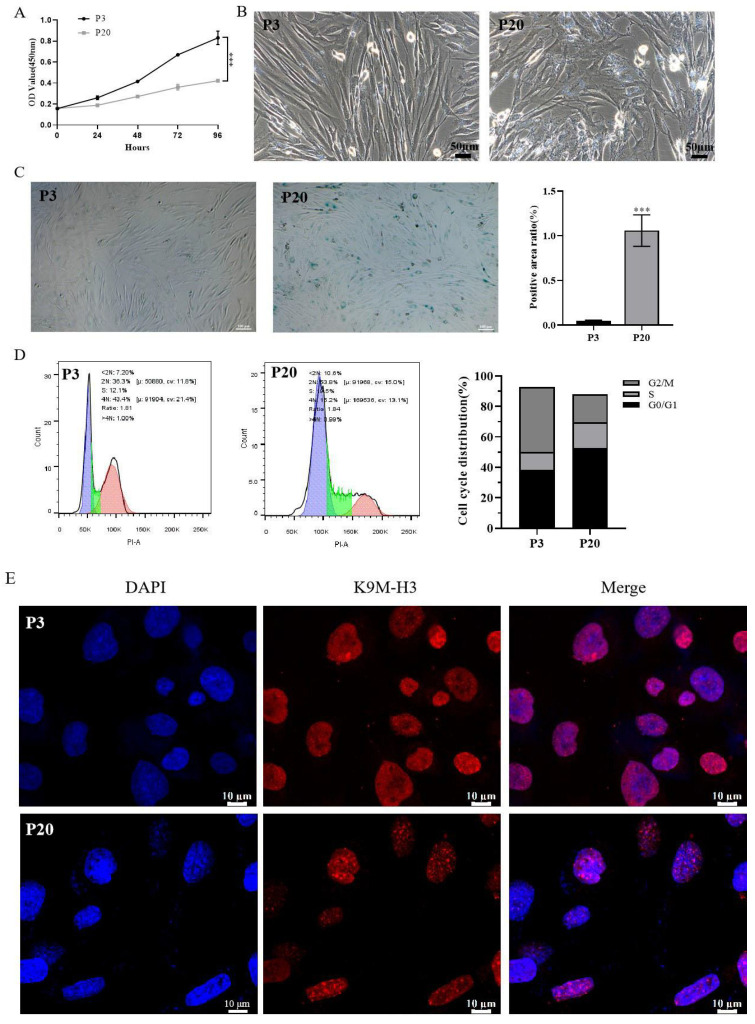
Passage-induced replicative senescence of HSFs. (**A**) Effects of passage number on growth curves of HSFs using a Cell Counting Kit-8 (CCK-8) assay. (**B**) Effects of passage number on cellular morphology of HSFs; scale bar is 50 µm. (**C**) Effects of passage number on senescence-associated β-galactosidase (SA-β-Gal) staining of HSFs; scale bar is 100 µm (left). Quantification is shown on the right. (**D**) Effects of passage number on cell cycle distribution of HSFs, as determined by flow-cytometric analysis (left). Blue, green, and red areas represent cells in G0/G1 phase, S phase, and G2/M phase, respectively. Quantification is shown on the right. (**E**) Effects of passage number on the formation of senescence-associated heterochromatin foci (SAHF) of HSFs, as photographed using a confocal microscope. The heterochromatin structure was mapped by K9M-H3 (red). Nuclei were stained with DAPI (blue) for counterstaining; scale bar is 10 µm; *** *p* < 0.001. The data shown are expressed as the mean ± standard deviation (SD) from three replicates.

**Figure 3 molecules-29-01468-f003:**
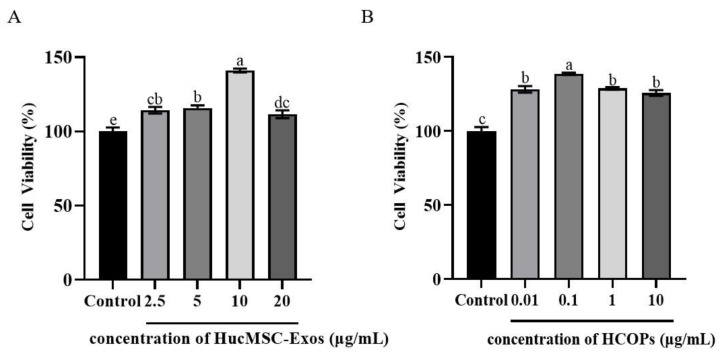
Effect of HucMSC-Exos (**A**) and hydrolyzed collagen oligopeptides (HCOPs) (**B**) at different concentrations on the proliferation of HSFs. Data were expressed as mean ± SD from three replicates. Different lowercase letters on bars indicate significant differences among samples treated with HucMSC-Exos or HCOPs at different concentrations (*p* < 0.05).

**Figure 4 molecules-29-01468-f004:**
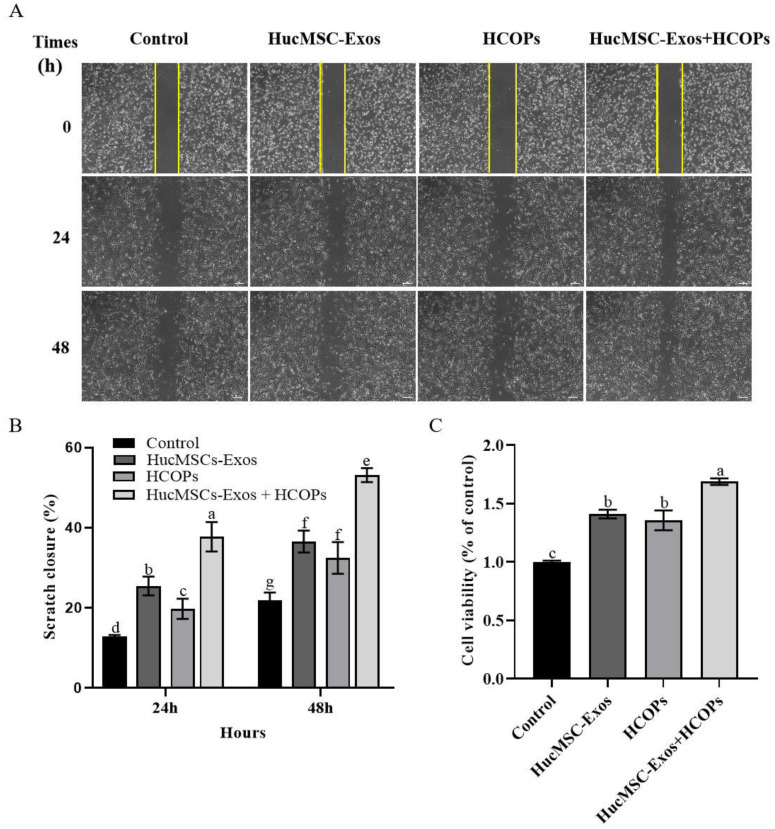
Effect of HucMSC-Exos, HCOPs, and HucMSC-Exos supplemented with HCOPs (HucMSC-Exos + HCOPs) on the proliferation and migration of HSFs. (**A**) Wound recovery rates of HSFs, modeled by cell-scratch assays; the area in the middle of the yellow lines represented the initial scratch area; scale bar is 200 µm. (**B**) The scratch closure rate is presented over time (*n* = 3). (**C**) HSF proliferation with the treatment of HucMSC-Exos, HCOPs, or HucMSC-Exos + HCOPs; *n* = 3. The data shown are expressed as mean ± SD. Different lowercase letters on bars indicate significant differences among samples (*p* < 0.05).

**Figure 5 molecules-29-01468-f005:**
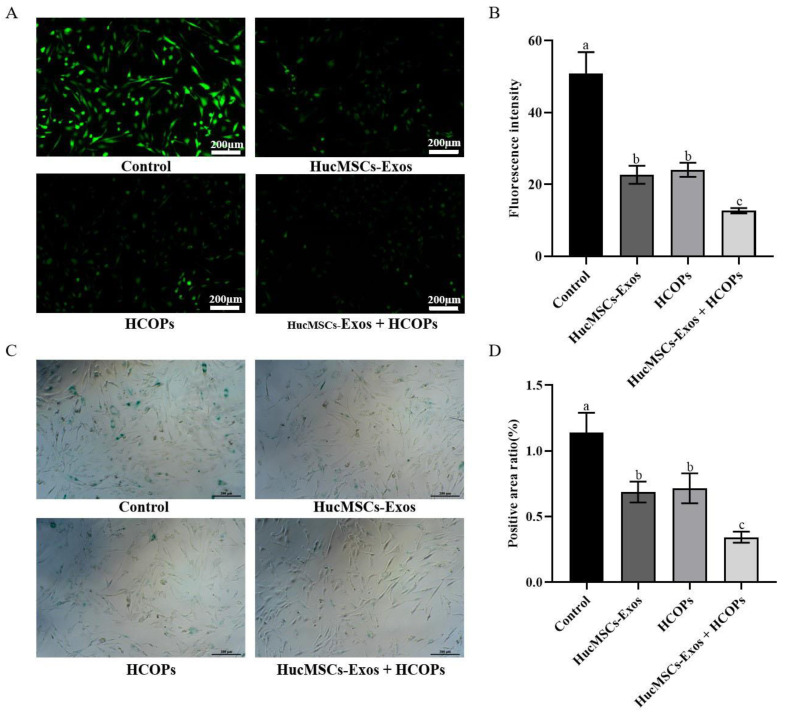
Effect of HucMSC-Exos, HCOPs, and HucMSC-Exos + HCOPs on the expression of reactive oxygen species (ROS) and SA-β-Gal activity in senescent HSFs. (**A**) ROS level of HSFs; scale bar is 200 µm. (**B**) Fluorescence intensity of ROS post-treatment with different samples. The data shown are expressed as the mean ± SD from three replicates. (**C**) The results of SA-β-Gal staining were observed after the co-culture of HSFs; scale bar is 200 µm. (**D**) Quantitative assays of SA-β-Gal staining. The data shown are expressed as mean ± SD from three replicates. Different lowercase letters on bars indicate significant differences among samples (*p* < 0.05).

**Figure 6 molecules-29-01468-f006:**
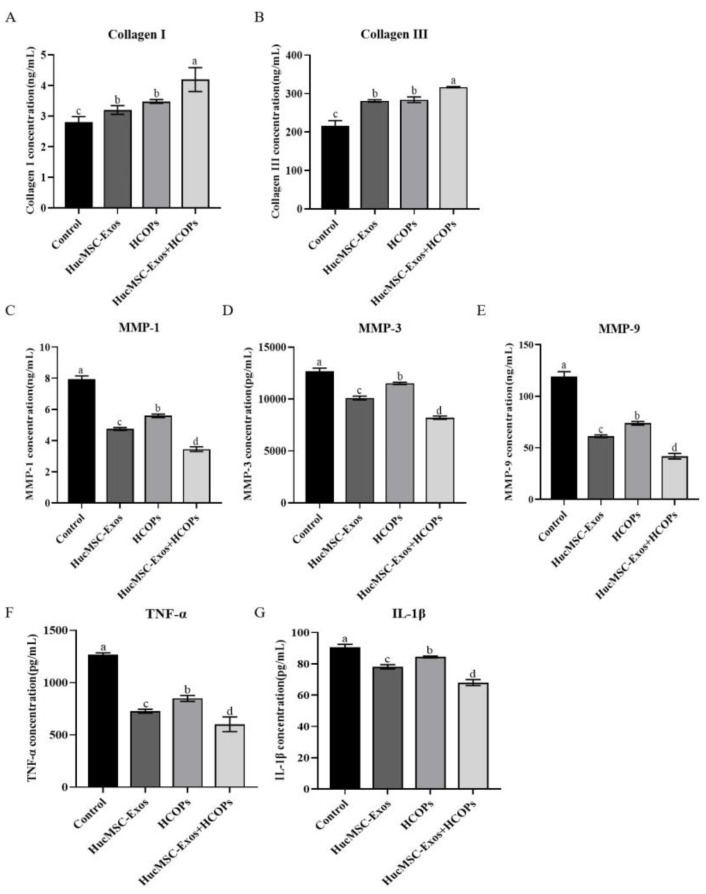
Effect of HucMSC-Exos, HCOPs, and HucMSC-Exos + HCOPs on ECM construction-related proteins and senescence-associated secretory phenotype (SASP) in HSFs. (**A**–**G**) Expression levels of collagen I and III, matrix-degrading metalloproteinases (MMP-1, MMP-3, MMP-9), tumor necrosis factor-alpha (TNF-α), and interleukin 1 beta (IL-1β) using an enzyme-linked immunosorbent assay (ELISA) kit (*n* = 3). The data shown are expressed as the mean ± SD. Different lowercase letters on bars indicate significant differences among samples (*p* < 0.05).

**Figure 7 molecules-29-01468-f007:**
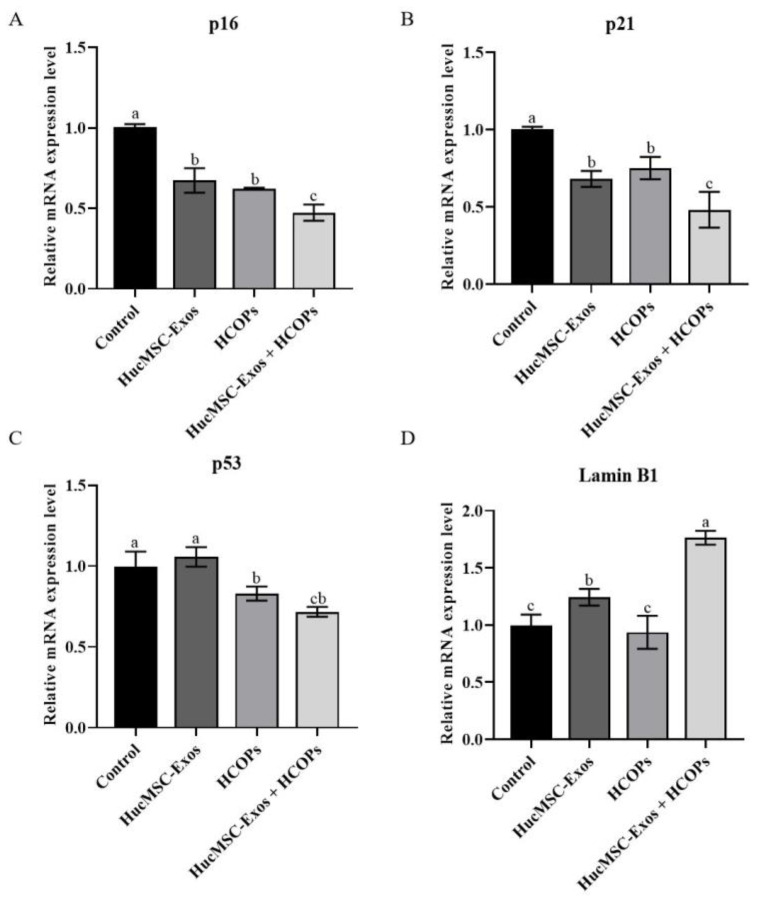
Effect of HucMSC-Exos, HCOPs, and HucMSC-Exos + HCOPs on the expression of p16, p21, p53, and lamin B1 gene in HSFs. The expression of each gene was normalized against the expression observed for the non-treated control group. (**A**–**D**) Relative messenger RNA expression levels of p16, p21, p53, and lamin B1. All data were expressed as mean ± SD from three replicates. Different lowercase letters on bars indicate significant differences among the samples tested (*p* < 0.05).

**Figure 8 molecules-29-01468-f008:**
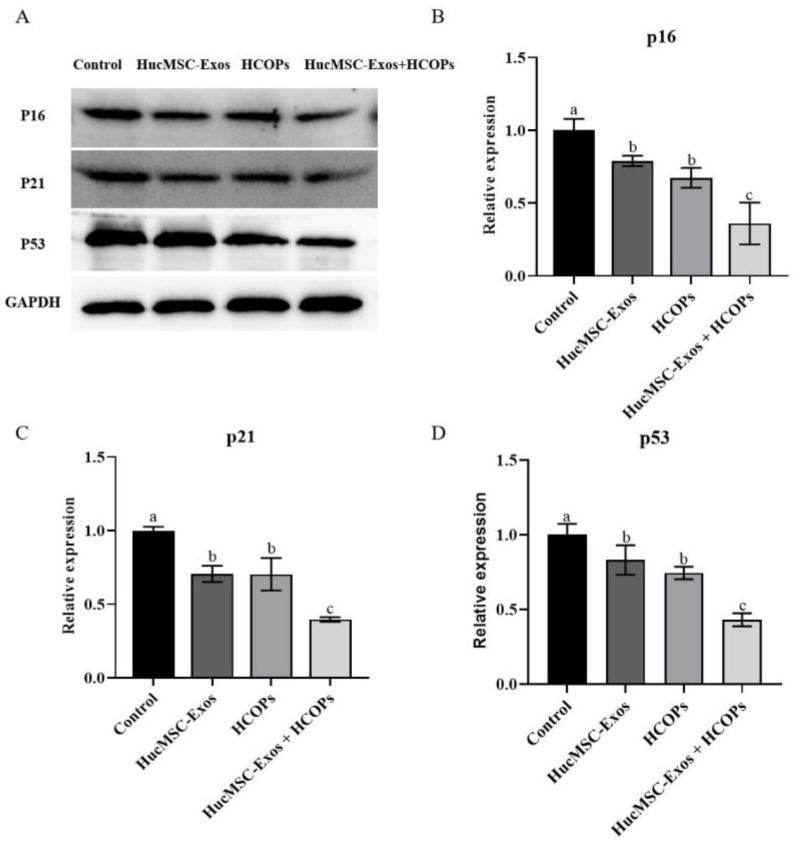
Effect of HucMSC-Exos, HCOPs, and HucMSC-Exos + HCOPs on the expression of p16, p21 and p53 proteins in HSFs. (**A**) Western blotting analysis of cells treated in different groups. (**B**−**D**) Quantification of p16, p21, and p53. The data shown were expressed as the mean ± SD (*n* = 3). Different lowercase letters on bars indicate significant differences among different samples (*p* < 0.05).

**Table 1 molecules-29-01468-t001:** Effect of exosomes derived from human umbilical cord mesenchymal stem cells (HucMSC-Exos), hydrolyzed collagen oligopeptides (HCOPs), and HucMSC-Exos + HCOPs on reactive oxygen species (ROS) expression and senescence-associated β-galactosidase (SA-β-Gal) activity in senescent human skin fibroblasts (HSFs).

Groups	ROS		SA-β-Gal	
	Fluorescence Intensity	Percentage of Control %	Positive Rate of SA-β-Gal	Percentage of Control %
Control	50.9 ± 6.0 a	100.0 ± 11.7 a	1.14 ± 0.15 a	100.0 ± 13.3 a
Exos	22.7 ± 2.5 b	44.6 ± 5.0 b	0.69 ± 0.08 b	60.3 ± 7.0 b
HCOPs	24.1 ± 2.0 b	47.3 ± 3.9 b	0.72 ± 0.11 b	62.8 ± 10.0 b
Exos + HCOPs	12.7 ± 0.7 c	24.9 ± 1.4 c	0.34 ± 0.04 c	30.1 ± 3.7 c

The data shown are expressed as the mean ± SD from three replicates. Different lowercase letters on bars indicate significant differences among samples (*p* < 0.05).

## Data Availability

Data are contained within the article and Appendix A.

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
