# Peer review of "The Combined Anti-Aging Effect of Hydrolyzed Collagen Oligopeptides and Exosomes Derived from Human Umbilical Cord Mesenchymal Stem Cells on Human Skin Fibroblasts"

_molecules, 2024, doi:10.3390/molecules29071468_

Round 1
Reviewer 1 Report
Comments and Suggestions for Authors
Since the exosome and HCOPs were added at the same time, the combine effect can’t be called enhancement. This title should be modified. Authors should discuss their respective mechanisms on viability, migration, senescence effect of HSF.
Author Response
- Since the exosome and HCOPs were added at the same time, the combine effect can’t be called enhancement. This title should be modified.
Response:
Thank you very much for your valuable advice. The title of the manuscript was revised as “The Combined Anti-aging Effect of Hydrolyzed Collagen Oligopeptides and Exosomes Derived from Human Umbilical Cord Mesenchymal Stem Cells on Human Skin Fibroblasts”.
- Authors should discuss their respective mechanisms on viability, migration,senescence effect of HSF.
Response:
Thank you very much for your valuable comments. To understand the anti-aging process, it is of great interest to explore the specific hidden molecular mechanism involved in it. The anti-aging effect and related mechanisms of exosomes have been reported in some literatures, but its clinical application is limited due to its yield and administration mode. In this manuscript, the effect of HSFs on cell viability, migration, and senescence have proven that combined use of exosomes with HCOPs have successfully enhanced the anti-aging effect on senescent cells, and thus had exerted a higher effect with a limited dose. In the next step our project, we will conduct a detailed study to check the effect of administration route on the anti-aging process at the animal level, and will explore the hidden molecular mechanism of action involved, so as to provide a theoretical basis for clinical application. However, the purpose of this study was to evaluate the combined effect of exosomes with HCOPs in anti-cellular senescence. If their respective mechanisms on viability, migration and senescence are to be explored, the manuscript will become longer and would deviate from the original intention of this study.
Reviewer 2 Report
Comments and Suggestions for Authors
The manuscript of Zhu and co-authors describes the effect of hydrolized collagen oligopetides together with exosomes derived from human umbilical cord MSC on senescent skin fibroblasts. This is an interesting topic concerning new possibilities in application of cell-free therapy in skin defects. However, there are some issues that need to be elucidated and corrected.
· In the whole text, including abstract, the Authors write about transforming growth factor beta and use the shortness IL-1β. Moreover, in Fig.6 G there is a histogram with a name IL-1β, whereas in the description to the figure there is again transforming growth factor beta. I really don’t know if the Authors performed the experiments regarding transforming growth factor beta (TGF β) or interleukin 1 beta (IL-1 β). Please correct it.
· Materials and Methods section:
1. point 4.8, lines 479-480. The Authors write “Three photographs per well were captured, and the cells …. were counted”. Which microscope was used for pictures?
2. point 4.10. lines 490-491. How were HSF prepared before fixation? The cells were grown on microscopic slides or cytospins were prepared?
3. point 4.11. Again, mixing up transforming growth factor beta (TGF β) and interleukin 1 beta (IL-1 β).
4. point 4.12. How much total RNA was transcribed into cDNA? In line 519-520 there is a mistake concerning number of cycles: as I understand it, 40 cycles are for three processes: denaturation, annealing and extension, not just denaturation at 95 degrees?
5. point 4.13. This point is treated very generally. There is no information on what amount of protein was applied to electrophoresis, what specific antibodies were used for WB (polyclonal, monoclonal, mouse, rat, etc.), which secondary antibodies were used, at what concentrations they were used, etc. There is also no information on how the post-blot quantification was done.
· Results:
1. I suggest that instead lowercase letters on bars (histograms) the Authors should mark asterisks connected with p value. Lowercase letters do not add anything to the figures, but illustrating statistical significance directly in the figure would be more legible, as it was illustrated in Figure 2 C.
2. Table 1, Table S3, Table S4 and Table S5 – I don’t understand the description Decrease % or Increase %. It is not the percentage by which a given value decreases or increases, but rather it is a percentage of control and should be described as such.
3. Figure 2 D – the description of cell cycle histogram of P16 is overlaid on the chart, both descriptions (P3 and P16) are difficult to read
4. Figure 6 G – is it really interleukin 1 beta (IL-1 β)?
· Discussion:
1. lines 306-309. I don’t understand whether this sentence refers to the MSC or the umbilical cord.
Comments on the Quality of English LanguageThe text contains numerous stylistic, punctuation and typographical errors. I strongly suggest that the text be proofread by an English native speaker.
Author Response
- In the whole text, including abstract, the Authors write about transforming growth factor beta and use the shortness IL-1β. Moreover, in Fig.6 G there is a histogram with a name IL-1β, whereas in the description to the figure there is again transforming growth factor beta. I really don’t know if the Authors performed the experiments regarding transforming growth factor beta (TGF β) or interleukin 1 beta (IL-1 β). Please correct it.
Response:
We are sorry and apologize for the wrong description which is just a written mistake and are highly obliged for highlighting the mistake. We have only performed experiments regarding interleukin 1 beta (IL-1 β) instead of transforming growth factor beta (TGF β) and the same is corrected in the revised manuscript.
- point 4.8, lines 479-480. The Authors write “Three photographs per well were captured, and the cells …. were counted”. Which microscope was used for pictures?
Response:
Three photographs per well were captured using an inverted microscope (Nikon, Ti2-U, Japan) and the same is added in the description of revised manuscript highlighted in yellow.
- point 4.10. lines 490-491. How were HSF prepared before fixation? The cells were grown on microscopic slides or cytospins were prepared?
Response:
HSFs cells were seeded (5 × 104 cells/well) into sterile cell climbing slices. After 80% confluency, HSFs were fixed with 4% paraformaldehyde and permeabilized with 0.5% Triton-X-100 solution at room temperature. We have described the relevant part in the revised manuscript.
- point 4.11. Again, mixing up transforming growth factor beta (TGF β) and interleukin 1 beta (IL-1 β).
Response:
We are sorry again for the wrong description about the interleukin 1 beta (IL-1 β). We have already corrected and highlighted the changed sentences in revised manuscript.
- point 4.12. How much total RNA was transcribed into cDNA? In line 519-520 there is a mistake concerning number of cycles: as I understand it, 40 cycles are for three processes: denaturation, annealing and extension, not just denaturation at 95 degrees?
Response:
Relative mRNA expression levels were normalized to GAPDH and calculated using the 2−∆∆Ct method with standard cycle conditions as: 95°C for 1 min, followed by 40 cycles of denaturation at 95°C for 15 s, annealing at 60°C for 15 s and extension at 72°C for 1 min. The same is corrected and highlighted in yellow in the revised manuscript.
In addition, 1 µg of the total RNA was transcribed into cDNA in this trial. We looked up the literatures and found that most of the literatures did not write this sentence in detail, so we did not write this part in our manuscript. I will add it to the manuscript if you feel that this is necessary.
- point 4.13. This point is treated very generally. There is no information on what amount of protein was applied to electrophoresis, what specific antibodies were used for WB (polyclonal, monoclonal, mouse, rat, etc.), which secondary antibodies were used, at what concentrations they were used, etc. There is also no information on how the post-blot quantification was done.
Response:
Proteins were separated using 10%-12% SDS-polyacrylamide gel with 20-30µg protein samples. The proteins were then transferred to polyvinylidene fluoride membranes (Millipore, USA). Blots were incubated with each primary antibody of target proteins (Rabbit polyclonal anti-CD81, Rabbit polyclonal anti-ALIX, Rabbit polyclonal anti-CDKN2A/p16-INK4a, Rabbit polyclonal anti-p21 Cip1, Mouse monoclonal anti-p53). Post-washed with TBST, the membrane was incubated again with a secondary antibody of goat anti-rabbit IgG and goat anti-mouse IgG conjugated to HRP (1:5000). Targeted protein bands were then scanned using an ImageQuant LAS 4000 Chemiluminescence Imager (GE, USA). Taking GAPDH as the reference protein, Image J software was used to analyze the gray value to quantify protein expression. The changes have been made in the revised manuscript and are highlighted in yellow.
- I suggest that instead lowercase letters on bars (histograms) the Authors should mark asterisks connected with p value. Lowercase letters do not add anything to the figures, but illustrating statistical significance directly in the figure would be more legible, as it was illustrated in Figure 2 C.
Response:
Thank you very much for your valuable suggestion. It is indeed very intuitive to mark significance with asterisks connected with p value, as it was illustrated in Figure 2 C. However, there are only two sets of data needed to be compared in Figure 2 C, while the rest of the data required pairwise comparisons between multiple groups of data. If we choose the asterisk connected with p value to mark significance, the picture would be complicated. So we chose the letter marking method, where the same letters indicate no statistical significance between data, while different letters indicate statistical significance between data. (p < 0.05).
- Table 1, Table S3, Table S4 and Table S5 – I don’t understand the description Decrease % or Increase %. It is not the percentage by which a given value decreases or increases, but rather it is a percentage of control and should be described as such.
Response:
Thanks for your valuable suggestion and we completely agreed with you. It should be described as “percentage of control” and we have already changed this part and highlighted it in yellow.
- Figure 2 D – the description of cell cycle histogram of P16 is overlaid on the chart, both descriptions (P3 and P16) are difficult to read
Response:
We are sorry for the mistake by labeing P20 as P16 in Figure 2D and is changed accordingly in the revised manuscript. In addition, literature review have shown that the cell cycle is often represented by overlaid histogram, therefore, we used overlaid histogram in our manuscript. However, we additionally added grouped histogram in the supplementary data (Figure S1), which will be replaced if the reviewer feels it necessary.
- Figure 6 G – is it really interleukin 1 beta (IL-1 β)?
Response:
We feel terribly sorry for the wrong description about the interleukin 1 beta (IL-1 β) and thank you very much for your kind comments. We have already replaced the relevant part in the revised manuscript and is highlighted in yellow.
- lines 306-309. I don’t understand whether this sentence refers to the MSC or the umbilical cord.
Response:
We feel terribly sorry for the wrong description about the human umbilical cord The sentence refers to the HucMSCs which is corrected and is highlighted in yellow.
- The text contains numerous stylistic, punctuation and typographical errors. I strongly suggest that the text be proofread by an English native speaker.
Response:
Thanks for your valuable suggestion. We have invited an English native speaker to proofread our manuscript, which have double checked the whole manuscript and have made necessary changes where needed as labeled in red letter.
Reviewer 3 Report
Comments and Suggestions for Authors
The study investigates the potential of hydrolyzed collagen oligopeptides (HCOPs) to enhance the anti-aging effects of exosomes derived from human umbilical cord mesenchymal stem cells (HucMSC-Exos) on senescent human skin fibroblasts, addressing the growing interest in leveraging regenerative medicine for skin aging interventions.
The abstract is well-written and concise. It effectively summarizes the research objectives, methods, and key findings.
The keywords are relevant and appropriately chosen, providing a clear indication of the study's focus.
The introduction provides a comprehensive background on skin aging, the role of fibroblasts, and the use of stem cells in anti-aging therapies. However, it would be beneficial to include more recent references to strengthen the foundation of the study. The research gap is well-defined, emphasizing the limitations of current stem cell therapies and the potential of exosomes. However, a clearer transition to the specific role of hydrolyzed collagen oligopeptides (HCOPs) in enhancing anti-aging effects could be provided.
Result: the exosome characterization is comprehensive, encompassing size distribution, TEM analysis, and immunoblotting, yet lacking information on exosome yield and concentration in the conditioned medium. The replicative senescence model is well-established, but further clarification on the rationale for choosing passage 20 is needed, addressing potential effects of passage number. Additionally, the justification for specific concentrations of HucMSC-Exos and HCOPs requires elaboration, with a discussion on their relevance to clinical applications for a more nuanced interpretation of results.
The discussion adeptly compares the effects of HucMSC-Exos, HCOPs, and their combination, yet requires a more profound analysis of how HCOPs enhance anti-aging effects, both mechanistically and in comparison to other studies, for a robust discussion. Further, a detailed exploration of mechanistic insights into how HCOPs contribute to observed effects, elucidating potential pathways and interactions with cellular processes, is warranted. The study's clinical relevance is addressed for in vitro effects, but a more thorough discussion on translating these findings into practical applications and acknowledging challenges would enhance the overall impact. While limitations are briefly acknowledged, a more comprehensive exploration of existing problems with exosome application and thoughtful suggestions for future research directions would enrich the discussion.
The study presents valuable insights into the potential synergy between HCOPs and HucMSC-Exos in anti-skin aging effects. Addressing the suggestions and enhancing the discussion with more mechanistic details would further strengthen the scientific impact of the research.
Comments on the Quality of English LanguageThe overall language and grammar are good. However, some sentences could be rephrased for better clarity and flow.
Author Response
- The introduction provides a comprehensive background on skin aging, the role of fibroblasts, and the use of stem cells in anti-aging therapies. However, it would be beneficial to include more recent references to strengthen the foundation of the study.
Thanks for your valuable suggestion. We have added more recent references in our manuscript.
- Takaya, K.; Kishi, K. Regulation of ENPP5, a senescence-associated secretory phenotype factor, prevents skin aging. Biogerontology 2024.
14. Wang, Z.; Ren, H.; Su, P.; Zhao, F.; Zhang, Q.; Huang, X.; He, C.; Wu, Q.; Wang, Z.; Ma, J. Adipose mesenchymal stem cell-derived exosomes promote skin wound healing in diabetic mice by regulating epidermal autophagy. Burns & Trauma 2024, 12, tkae001.
- Zhang, X.; Ding, P.; Chen, Y.; Lin, Z.; Zhao, X.; Xie, H. Human umbilical cord mesenchymal stem cell‐derived exosomes combined with gelatin methacryloyl hydrogel to promote fractional laser injury wound healing. Int Wound J2023, 20, 4040-4049.
- Nguyen, D.D.N.; Vu, D.M.; Vo, N.; Tran, N.H.B.; Ho, D.T.K.; Nguyen, T.; Nguyen, T.A.; Nguyen, H.N.; Tu, L.N. Skin rejuvenation and photoaging protection using adipose‐derived stem cell extracellular vesicles loaded with exogenous cargos. Skin Res Technol2024, 30.
- Yang, D.; Liu, Q.; Xu, Q.; Zheng, L.; Zhang, S.; Lu, S.; Xiao, G.; Zhao, M. Effects of collagen hydrolysates on UV-induced photoaging mice: Gly-Pro-Hyp as a potent anti-photoaging peptide. Food & Function 2024.
- Pinto, H.; Sánchez-Vizcaíno Mengual, E. Exosomes in the Real World of Medical Aesthetics: A Review. Aesth Plast Surg 2024.
- The research gap is well-defined, emphasizing the limitations of current stem cell therapies and the potential of exosomes. However, a clearer transition to the specific role of hydrolyzed collagen oligopeptides (HCOPs) in enhancing anti-aging effects could be provided.
Response:
Thanks for your valuable suggestion. Numerous studies have recently shown the therapeutic role of MSCs derived exosomes to regenerate skin and enhance various renal, cardiovascular, and liver injuries, as well as have therapeutic potential in anti-skin aging. However, low yield, difficult preparation, and harsh transportation and storage conditions lead to the partial degradation of active components in exosomes and thus decline their anti-aging ability to limit the wide application of exosomes. In some studies, exosomes are combined with drugs or specific therapeutic components to improve disease treatment, providing a method to enhance the applicative value of exosomes.
Therefore, after preliminary screening, we compared the efficacy of typical anti-aging drugs such as collagen peptide, ascorbic acid, VB6, rice protein, rapamycin, and resveratrol, and the possible effect of various combinations of these drugs with exosomes was investigated. Based on the pre-experimental results, the most therapeutically promising HCOPs were finally identified as the combined component of exosomes with HCOPs, which was selected for further investigation of their anti-aging efficacy. Hydrolyzed collagen oligopeptides (HCOPs) are enzyme hydrolytic forms of collagen, comprising of a popular ingredient which is supposed to be an antioxidant, and thus have skin anti-aging effects. We herein evaluated the combined anti-aging effects of HCOPs and exosomes derived from human umbilical cord mesenchymal stem cells (HucMSC-Exos) using a senescence model established in human skin fibroblasts (HSFs). According to your kind advice, we have already changed the relevant part as highlighted in yellow.
- Result: the exosome characterization is comprehensive, encompassing size distribution, TEM analysis, and immunoblotting, yet lacking information on exosome yield and concentration in the conditioned medium.
Response:
We highly appreciate the valuable suugestion of lacking information on exosome yield and concentration in the conditioned medium. BCA assay showed protein concentration of HucMSC-Exos obtained from 100 mL HucMSC-CM which was 2149 µg/mL. The relevant part has been added in the revised manuscript and highlighted in yellow.
- The replicative senescence model is well-established, but further clarification on the rationale for choosing passage 20 is needed, addressing potential effects of passage number.
Response:
Thank you very much for your kind comments. Actually, we observed the changes of HSFs cells during multiple passages through cellular proliferation and morphology, cell cycle kinetics, SA-β-Gal activity, and SAHF experiments. The results confirmed that P20 cells showed obvious senescence characteristics, and therefore were selected for further study. In fact, literature review has also shown that cells from P30-P45 passages were used as a senescent cell model. The purpose of this study was to investigate the combined anti-aging effect of exosomes and HCOPs, and the experimental results have proven our hypothesis. As for the cells after P20 passages, we found that there were no significant difference between the cells within three passages, and thus did not studied the more aging cells, because the cell growth rate was significantly slower and the replicative senescence model construction time was long, we did not have more time to verify the further effect of cell generation. However, the results positively proved that combination of exosomes and HCOPs does have a synergistic anti-aging effect, and it is believed that the same trend is still suitable for the cells after the P20 generations.
- Additionally, the justification for specific concentrations of HucMSC-Exos and HCOPs requires elaboration, with a discussion on their relevance to clinical applications for a more nuanced interpretation of results.
Response:
Thank you very much for your kind comments. According to the literature, it is found that the concentrations range of the exosomes on cells is between 2 - 200 µg/mL, and the commonly used concentrations range is between 5 - 20 µg/mL. The cell proliferation assay in our study found that the exosomes has the best effect in 10 µg/mL on the model of replicative senescence of HSFs, thus this concentration was selected for study. There are few studies on the effect of collagen peptides on cells. Moreover, different studies have different sources of collagen peptides, different preparation methods, and different compositions, leading to their different anti-aging effects on cells, and there is no specific concentrations range to refer to. The results of the cell proliferation assay in our study found that the best effect of HCOPs was 0.1 µg/mL. In clinical application, exosemes is mostly used in injection, while collagen peptides is mostly used in oral administration. However, the cell experiments in vitro in our study provided some theoretical reference for its clinical use.
- The discussion adeptly compares the effects of HucMSC-Exos, HCOPs, and their combination, yet requires a more profound analysis of how HCOPs enhance anti-aging effects, both mechanistically and in comparison to other studies, for a robust discussion. Further, a detailed exploration of mechanistic insights into how HCOPs contribute to observed effects, elucidating potential pathways and interactions with cellular processes, is warranted. The study's clinical relevance is addressed for in vitro effects, but a more thorough discussion on translating these findings into practical applications and acknowledging challenges would enhance the overall impact. While limitations are briefly acknowledged, a more comprehensive exploration of existing problems with exosome application and thoughtful suggestions for future research directions would enrich the discussion.
Response:
Thank you very much for your valuable comments. It is of great interest to explore the hidden molecular mechanism of effect. The anti-aging effect and related mechanisms of exosomes have been reported in some literatures, but its practical clinical application is limited due to the yield and administration mode. In this manuscript, the results of cell viability, migration, and senescence effect of HSFs proved that the anti-aging effect of the exosomes on senescent cells is enhanced after the addition of HCOPs, and thus can exert a higher effect with a limited dose. In the next step, we will conduct a detailed study on the anti-aging effect and the way of administration at the animal level, and explore the specific mechanism of action, so as to provide a theoretical basis for clinical application.
- The overall language and grammar are good. However, some sentences could be rephrased for better clarity and flow.
Response:
Thanks for your valuable suggestion. We have invited an English native speaker to proofread our manuscript, which have double checked the whole manuscript and have made necessary changes where needed as labeled in red letter.